# Assessing Multi-Site rs-fMRI-Based Connectomic Harmonization Using Information Theory

**DOI:** 10.3390/brainsci12091219

**Published:** 2022-09-09

**Authors:** Facundo Roffet, Claudio Delrieux, Gustavo Patow

**Affiliations:** 1Department of Electrical and Computer Engineering (DIEC), Universidad Nacional del Sur, Bahía Blanca AR-B8000, Argentina; 2Department of Electrical and Computer Engineering (DIEC), Universidad Nacional del Sur and National Council for Scientific and Technical Research (CONICET), Bahía Blanca AR-B8000, Argentina; 3ViRVIG, University of Girona, 17003 Girona, Spain

**Keywords:** rs-fMRI, harmonization, information theory, neuroscience, multi-site acquisition

## Abstract

Several harmonization techniques have recently been proposed for connectomics/networks derived from resting-state functional magnetic resonance imaging (rs-fMRI) acquired at multiple sites. These techniques have the objective of mitigating site-specific biases that complicate its subsequent analysis and, therefore, compromise the quality of the results when these images are analyzed together. Thus, harmonization is indispensable when large cohorts are required in which the data obtained must be independent of the particular condition of each resonator, its make and model, its calibration, and other features or artifacts that may affect the significance of the acquisition. To date, no assessment of the actual efficacy of these harmonization techniques has been proposed. In this work, we apply recently introduced Information Theory tools to analyze the effectiveness of these techniques, developing a methodology that allows us to compare different harmonization models. We demonstrate the usefulness of this methodology by applying it to some of the most widespread harmonization frameworks and datasets. As a result, we are able to show that some of these techniques are indeed ineffective since the acquisition site can still be determined from the fMRI data after the processing.

## 1. Introduction

Magnetic resonance imaging (MRI) is an imaging modality that allows, among other things, the monitoring and sensing of the neuronal activity in the brain. Several drivers are raising the application horizons of MRI imaging. To mention the main ones, MRI is innocuous and non-invasive, resonators are steadily becoming less expensive, and with increasing capabilities, the knowledge of the human brain’s anatomy and physiology is turning more definite and precise, and new analysis techniques are able to extract subtle, latent information that is fundamental to several research and clinical purposes. In particular, resting-state functional magnetic resonance imaging (rs-fMRI) is able to capture interactions between brain regions that, in turn, can lead to evaluating several biomarkers of interest. Currently, acquisitions obtained from rs-fMRI data make it possible to thoroughly study several aspects of the human brain function, both in healthy subjects and those diagnosed with neurological or even psychiatric conditions, including Alzheimer’s disease [1,2], schizophrenia [3,4], and autism spectrum disorder [5,6]. In order to carry out these studies correctly and obtain an adequate statistical significance, it is necessary to have a high number of acquisitions, which is not always possible in a single site and session. Therefore, the number of studies using images acquired at multiple sites has been increasing over the years [7,8,9,10]. This allows the speeding up the data collection and analysis process, and increasing the sample size naturally leads to greater predictive power and more sophisticated studies. Multi-site acquisition is also critically important to be able to generalize the analysis results or when investigating low prevalence or rare disorders [11,12].

However, multi-site data studies often introduce unwanted acquisition heterogeneities or artifacts that can greatly impact the significance of the results derived from them [13,14,15,16]. This variability (which is not biological or anatomical in origin) is due to various causes that are difficult or impossible to control, including electromechanical differences between different equipment, manufacturers, and models, different arrangements of magnetic field density and signals of radiofrequency (even among similar equipment), the variation in the sizes of the voxels, the parameters and acquisition sequences of the scanners, the calibration carried out, etc. [17,18,19]. For these reasons, the generation of harmonization techniques between different sites is a subject that is attracting an ever-growing interest within neuroimaging. In the last decade, a large number of methods have been developed [20] capable of applying mathematical or statistical concepts to reduce unwanted variabilities produced during acquisition without compromising the quality of the anatomical or physiological information obtained. Among these methods, *ComBat* [21,22,23,24] stands out, having great popularity thanks to its good results since its adaptation from the field of genomics to neuroscience in 2017 [25,26]. Since then, new techniques based on and/or inspired by ComBat have emerged with the aim of further improving the ability to generalize the results obtained [27,28,29,30,31].

However, other perspectives on the full understanding of human brain functioning appear to be mandatory. In particular, the structure and dynamics of graphs and networks in general (from social to neural networks, on a wide spectrum of scales) are of paramount importance in many scientific fields in which the collective behavior of a network of entities is relevant. Neuroscience does not escape this paradigm since it is possible to derive neural graphs (called *connectomes*) from the information extracted from rs-fMRI data, among other modalities. Having high-quality connectome information enables a significant amount of studies related to mental health and neurocognitive conditions. Recently, some methods derived from *Information Theory* and *Statistical Complexity* began to be applied in this field. In particular, Amico et al. [32] introduced a framework combining two different theoretical information measurements for fixed structural topology systems over which different communication processes are taking place, with the aim of exploring communication dynamics in large-scale brain networks. Similarly, Luppi and Stamatakis [33] combined network topology and Information Theory to construct representative brain networks. In spite of these successful recent examples, the topic is still open to further research, so we recommend the interested reader to refer to the thesis of Bonmati [34], the book by Piasini and Panzeri [35], or the review by Farahani et al. [36] for further insights into this topic. In other contexts, these tools have been recognized for their ability to create a robust characterization of complex phenomena, especially in time series analysis (e.g., biosignals, cryptocurrencies, etc.) [37,38]. However, the analysis of complex networks through a characterization within this theoretical framework (for example, *Shannon Entropy*, *Statistical Complexity* and *Fisher Information* [39]) has not yet been applied to connectomics.

In this paper, we present the use of Network Entropy and Network Fisher Information (and the Shannon–Fisher causal plane) to assess the robustness of several rs-fMRI harmonization techniques. Specifically, studies were performed on four multi-site datasets under four different processing approaches: unharmonized, applying the ComBat method [21], applying the CovBat method [28], and applying the traveling-subject method [31]. The results show that, depending on the method used, the harmonization is not entirely successful since our techniques continue to be able to determine the acquisition sites after the harmonization was applied, which shows the existence of site-related biases that are not completely mitigated. These results demonstrate the relevance and applicability of Statistical Complexity and Information Theory to rs-fMRI data analysis, and (to the best of our knowledge) the contribution presented here would be the first application of this theoretical framework in brain connectomics. See Figure 1.

## 2. Materials and Methods

### 2.1. Datasets

In this section, we describe the different datasets used for testing. Since the aim is to compare the sites within each multi-site dataset, we can see that it does not matter that each downloaded dataset had different processing steps already applied. Further, it is worth noting that three of the datasets are given as sets of time series (IMPAC, ABIDE, and ADHD-200), while one was given as a set of connectivity matrices of Pearson correlation values (SRPBS). See below.

#### 2.1.1. IMPAC

The *IMaging-PsychiAtry Challenge* [40] dataset contains rs-fMRI scans from 34 different sites of 1127 subjects, of which 537 are diagnosed with autism spectrum disorder, and the rest are healthy control subjects. The signals extracted from the images are based on seven different atlases: MSDL (39 ROIs), Harvard-Oxford (48 ROIs), Basc064 (64 ROIs), Basc122 (122 ROIs), Basc197 (197 ROIs), Craddock (249 ROIs), and Power (264 ROIs). The code used for extracting the time series can be found at https://github.com/ramp-kits/autism/blob/master/preprocessing/extract_time_series.py, (accessed on 15 April 2022 ). The imaging parameters are not available for this dataset, but instructions about how to obtain the repetition times can be found at https://github.com/ramp-kits/autism/issues/36, (accessed on 15 April 2022).

#### 2.1.2. ABIDE

*ABIDE* [41] is a publicly accessible repository of 20-site rs-fMRI data from 17 different international institutions. The dataset is composed of scans of 539 individuals diagnosed with autism spectrum disorder and 573 healthy control individuals. However, because some data had downloading problems, the number of total subjects was reduced from 1112 to 884. The image acquisition parameters can be found at https://fcon_1000.projects.nitrc.org/indi/abide/, (accessed on 15 April 2022). The data were preprocessed using the standard pipeline *DPARSF* (Data Processing Assistant for Resting-State fMRI Toolbox), which is based on the parametric statistical mapping. Finally, the time series were extracted following the definitions of seven different atlases: Talaraich–Tournoux (97 ROIs), Harvard-Oxford (111 ROIs), AAL (116 ROIs), Eickhoff–Zilles (116 ROIs), Dosenbach (161 ROIs), Craddock (200 ROIs), and Craddock (400 ROIs). Details of the atlases used and the pipeline can be found at http://preprocessed-connectomes-project.org/abide/Pipelines.html, (accessed on 15 April 2022), while the scan parameters for each site are available at http://fcon_1000.projects.nitrc.org/indi/abide/abide_I.html, (accessed on 15 April 2022).

#### 2.1.3. ADHD-200

As part of the International Neuroimaging Datasharing Initiative (INDI), the *ADHD-200* dataset [42] is a collaboration of 8 imaging sites, composed of neuroimaging data from 362 children and adolescents diagnosed with Attention Deficit Hyperactivity Disorder (ADHD) and 585 typically developing controls (total 947 subjects). As some scans had downloading problems or missing data, the number of subjects used in this work was reduced to a total of 768 subjects. The data were preprocessed using the Athena pipeline, which is based on tools from the AFNI and FSL software packages. The time series extracted from the images are based on six different atlases: Talaraich–Tournoux (97 ROIs), Harvard-Oxford (111 ROIs), AAL (116 ROIs), Eickhoff–Zilles (116 ROIs), Craddock (190 ROIs), and Craddock (351 ROIs). The data can be found at https://www.nitrc.org/frs/?group_id=383, (accessed on 15 April 2022), and details of the pipeline at https://www.nitrc.org/plugins/mwiki/index.php/neurobureau:AthenaPipeline, (accessed on 15 April 2022). The imaging parameters are detailed at http://fcon_1000.projects.nitrc.org/indi/adhd200/, (accessed on 15 April 2022).

#### 2.1.4. SRPBS

This dataset [43,44] includes data from subjects with four different diagnoses and healthy control subjects who were examined at nine sites corresponding to eight institutions. Of the 805 participants, 482 are healthy, 161 have a major depressive disorder, 49 have autism spectrum disorder, 65 have obsessive-compulsive disorder, and 48 have schizophrenia. Each participant underwent a single session of rs-fMRI for 5 to 10 min. The time series extraction procedure is detailed in the work by Yamashita and co-authors [31], where 268 regions of interest were delineated. Of note, participants who reported high levels of head movement were excluded, resulting in a reduction in the size of the dataset to 637 subjects. It is worth mentioning that this dataset also includes traveling-subject data, and it thus could be applicable to the traveling-subject harmonization methodology. The data are available at https://bicr.atr.jp/dcn/en/download/harmonization/, (accessed on 15 April 2022), while scan parameters can be found at https://bicr.atr.jp/rs-fmri-protocol-2/, (accessed on 15 April 2022).

#### 2.1.5. Traveling-Subject Dataset

As will be explained in a later section, the traveling-subject dataset [31] is necessary to estimate measurement bias across sites in the SRPBS dataset. It is composed of data of 9 healthy participants that were scanned at each of 12 sites, which included the 9 sites in the SRPBS dataset, producing a total of 411 scan sessions.

### 2.2. Connectivity

We started by applying a bandpass filter of 0.04–0.07 Hz to the BOLD time series to select adequate low frequencies. Then, the instantaneous phases $\phi_{i}\left(t\right)$ of each region *i* were estimated by applying the Hilbert transform to the filtered signals. The phase coherence $P_{ij}\left(t\right)$ between two regions *i* and *j* at a time *t* was calculated using the cosine of the phase difference, as shown in Equation (Equation 1).
(1)$$P_{ij}\left(t\right)=\cos\left(\phi_{i}\left(t\right)-\phi_{j}\left(t\right)\right).$$

Since the Hilbert transform expresses signals in polar coordinates, using the cosine function makes two regions have a phase coherence close to 1 when their time series are in phase, 0 when they are orthogonal, and −1 when they are out of phase. In this way, the phase interaction matrix P(t) represents the instantaneous phase synchrony between each of the regions [45,46]. This procedure results in a matrix of size N×N×T for each subject, where *N* is the number of regions and *T* is the total number of points in the time series:(2)$$\langle P\rangle =\sum_{t=0}^{\infty }\frac{P(t)}{T}.$$

Since *T* is a different value for each subject, the temporal dimension was eliminated by averaging each of the matrices over time. The procedure detailed above was applied to the IMPAC, ABIDE, and ADHD-200 datasets. For the SRPBS dataset, on the other hand, we worked with correlation matrices since the data provided by the dataset correspond to the Pearson correlation coefficient’s values are widely used in previous studies.

### 2.3. Harmonization

As introduced in the previous section, to reduce potential biases and non-biological variances introduced by different acquisition sites and scanners, there are various methods of data harmonization. In this work, the effectiveness of three of them is evaluated. One is ComBat [21,22,23], which is probably the most accepted and widely used in the literature. The second is CovBat [28], which is a refined and improved version of ComBat. The last one is the traveling-subject method, whose authors have shown that it can be more effective than ComBat in certain cases [31]. Below we briefly describe their main features. It is important to emphasize that in this work, for both ComBat and CovBat, the biological covariates to be protected during the removal of scanner/site effects were defined as the gender, age, and diagnosis of each of the subjects.

#### 2.3.1. ComBat

The *ComBat* technique, originally created to be used in Genomics analysis, is perhaps the most commonly used for the harmonization of brain connectivity data. It is based on Bayes’ empirical method, assuming that errors in the data can be corrected by adjusting the means and variances of the different acquisition sites. It has been shown to be able to eliminate site differences while adequately maintaining biological variability [24,26]. Defining $y_{tjv}$ as the evaluation at site *t*, participant *j*. and characteristic *v*, the ComBat regression model can be written as
(3)$$y_{tjv}=\alpha_{v}+X_{tj}\beta_{v}+\gamma_{tv}+\delta_{tv}\epsilon_{tjv},$$
where $\alpha_{v}$ is the average connectivity of the feature *v*, $X_{tj}$ is the design matrix for the covariances of interest, and $\beta_{v}$ is the regression vector of coefficients corresponding to *X*. In turn, $\gamma_{tv}$ and $\delta_{tv}$ represent the additive and multiplicative terms of the site *i* for the feature *v*, respectively, while $\epsilon_{tjv}$ represents an error term assumed to arise from a normal distribution with zero mean and variance $\sigma_{v}^{2}$. The values harmonized by ComBat are then defined as:(4)$$y_{tjv}^{ComBat}=\frac{y_{tjv}-\hat{\alpha_{v}}-X_{tj}\hat{\beta_{v}}-\gamma_{tv}^{*}}{\delta_{tv}^{*}}+\hat{\alpha_{v}}+X_{tj}\hat{\beta_{v}},$$
where $\gamma_{tv}^{*}$ and $\delta_{tv}^{*}$ are the empirical Bayesian estimates of the parameters $\gamma_{tv}$ and $\delta_{tv}$. Therefore, biological and non-biological terms are modeled and estimated to algebraically eliminate the additive and multiplicative effects of the sites. The calculations were made using the library available at https://github.com/Jfortin1/ComBatHarmonization, (accessed on 15 April 2022).

#### 2.3.2. CovBat

The method called *Correcting Covariance Batch Effects* (*CovBat*) was proposed to remove site effects in mean, variance, and covariance. It was built on top of the ComBat framework, assuming that the features follow Equation (Equation 3). However, the error vectors $\epsilon_{tjv}$ may be spatially correlated and differ in covariance across sites, so this method modifies principal component scores to shift each within-site covariance to the pooled covariance structure. This means that the first term of Equation (Equation 4) is assumed to have a mean of 0, but its covariance matrix Σ may differ between sites. Therefore, principal component analysis (PCA) is performed to obtain an estimation of the eigenvalues λ and eigenvectors ϕ of Σ. The principal component scores are defined as
(5)$$\xi_{tjk}=\mu_{tk}+\rho_{tk}\epsilon_{tjk},$$
where $\epsilon_{tjk}$ is a zero-mean normal distribution and $\mu_{tk}$ and $\rho_{tk}$ are the center and scale parameters corresponding to principal components k=1,2,⋯,K where *K* is a hyperparameter selected to capture the desired proportion of the variation in the observations. The parameters are estimated by finding the values that bring each site’s mean and variance in scores to the pooled mean and variance. Then, the site effects are removed via
(6)$$\xi_{tjk}^{CovBat}=\frac{\xi_{tjk}-\hat{\mu_{tk}}}{\hat{\rho_{tk}}}.$$
The CovBat-adjusted residuals are obtained as
(7)$$e_{tj}^{CovBat}=\sum_{k=1}^{K}\xi_{tjk}^{CovBat}\hat{\phi_{k}}+\sum_{l=K+1}^{q}\xi_{tjl}\hat{\phi_{l}}.$$
Adding the intercepts and covariates’ effects, the harmonized values result in
(8)$$y_{tjv}^{CovBat}=e_{tjv}^{CovBat}+\hat{\alpha_{v}}+X_{tj}\hat{\beta_{v}}.$$

#### 2.3.3. Traveling-Subject Method

This method is based on the identification of measurement biases, sampling biases, disorder factors, and subject factors. The measurement bias *m* for each site is defined as the deviation of the connectivity value between each pair of regions of interest from its average over all sites and is due to the differences between the properties of the scanners involved. The sampling bias *s*, introduced due to differences in participant groups between sites, is assumed to be different for subjects diagnosed with different disorders. Disorder factors *d* are defined as deviations from control subjects. In turn, the factors of the subjects *p* are calculated as the deviation of the connectivity from the average of the participants.

All the biases and factors mentioned above are estimated by fitting a linear regression model using ordinary least squares with $L^{2}$ regularization. The connectivity value *v* for subject *j* is then given by
(9)$$y_{jv}=x_{j}^{m}m_{v}+x_{j}^{s}s_{v}+x_{j}^{d}d_{v}+x_{j}^{p}p_{v}+const+e,$$
where const represents the average connectivity between all participants and *e* the noise. The vectors $x^{m}$, $x^{s}$, $x^{d}$, and $x^{p}$ are one-hot encoded. It is difficult to separate differences between sites using a single dataset because the two types of biases defined are correlated across sites. Therefore, in order to use this method, it is necessary to have an extra dataset (the so-called traveling-subject dataset), where the participants are constant; that is, it must be composed of scans of a constant set of healthy subjects in each of the sites in the original dataset. This is why the measurement bias is estimated only using the traveling-subject dataset. It is precisely the acquisition of this traveling-subject dataset that presents the main problem of the technique, since it requires that the same reference subjects physically travel to all the acquisition sites, with the consequent logistic problem that this implies. Finally, matching is achieved by subtracting the estimated measurement biases $\hat{m_{v}}$ from the connectivity values, resulting in
(10)$$y_{jv}^{Traveling}=y_{jv}-x_{j}^{m}\hat{m_{v}}.$$

### 2.4. Assessment of Harmonization Quality

In this subsection, we will introduce the specific theoretic information foundations that we use in this work. As already mentioned, the measures and methods derived from this background are capable of assessing the actual effectiveness of the different harmonization measures. Readers familiar with Information Theory, Shannon entropy, Fisher Information, and the causality-complexity plane can skip the reading of this subsection.

#### 2.4.1. Information Theory Measures

A key aspect of Information Theory is the concept of entropy as a measure of the uncertainty involved in the outcome of a random variable or process. These outcomes, in turn, are related to the probabilities (or relative frequencies) of the possible values that the variable or process may hold. Then, as the first step in our application of Information Theory in the context of rs-fMRI, it is necessary to define the probability distribution for the data. Our strategy was to binarize both the averaged phase interaction matrices and the correlation matrices using a threshold value of 0.5. Any other nontrivial threshold can be applied, and experiments show that the results to be exposed below are robust with respect to this choice in a wide range (*f.e.*, from 0.2 to 0.8). Connections with values greater than the threshold were assigned to 1, while the others were assigned to 0. In this way, each of the new matrices results in an adjacency matrix **A** that represents the neural graph of a subject model of the given threshold. Once obtained, the probability that a random walk goes from a node *i* of a graph to any other node *j* is calculated. This probability distribution $p_{i\to j}$ is defined for each node as
(11)$$p_{i\to j}=\left\{\begin{array}{cc} 0, & a_{ij}=0\\ 1/k_{i}, & a_{ij}=1 \end{array}\right.$$
where $k_{i}$ is the degree of the node, and its value is obtained as $k_{i}=\sum_{j}a_{ij}$.

#### 2.4.2. Shannon Entropy

Based on the $P^{\left(i\right)}$ distribution, the Shannon Entropy for each node can be defined as
(12)$$S\left[P^{\left(i\right)}\right]=-\sum_{j=1}^{N-1}p_{i\to j}\ln\left(p_{i\to j}\right),$$
where $P^{\left(i\right)}=\left\{p_{i\to j}:j=1,\ldots ,N\right\}$ is the probability distribution vector associated to node *i*. In turn, the Normalized Nodal Entropy for node *i* is obtained as
(13)$$H^{\left(i\right)}=\frac{S[P^{\left(i\right)}]}{\ln(N-1)}.$$
Finally, the Normalized Network Shannon Entropy (SE) is calculated by averaging the Normalized Nodal Entropy over the entire network, resulting in
(14)$$H=\frac{1}{N}\sum_{i=1}^{N}H^{\left(i\right)}.$$
SE is a global disorder measure commonly used in various applications of Information Theory. An advantage of SE in networks is that it is relatively insensitive to substantial changes in distributions that are concentrated in a small region of space. Therefore, it is able to quantify the heterogeneity of networks: H→0 for sparse networks and H→1 for fully connected networks.

#### 2.4.3. Fisher Information

Fisher Information is a statistical model aimed at measuring how much information about an unknown parameter can be obtained from a sample. In other words, it assesses the amount of information that an observable random variable within a population carries about an unknown parameter of the distribution that models the population. Using the notation of the previous subsection, the Normalized Fisher Information for a node *i* is given by
(15)$$F^{\left(i\right)}\left[P^{\left(i\right)}\right]=\frac{1}{2}\sum_{j=1}^{N-1}{\left[\sqrt{p_{i\to j+1}}-\sqrt{p_{i\to j}}\right]}^{2}.$$
Then, the Normalized Network Fisher Information (FI) is defined as
(16)$$F=\frac{1}{N}\sum_{i}F^{\left(i\right)}\left[P^{\left(i\right)}\right].$$
This measure can be interpreted in various ways, for instance, the ability to precisely estimate a parameter, the amount of information that can be extracted from a set of measures, or the state of disorder of a system. Unlike Shannon Entropy, Fisher Information is a local measure based on the gradient of the underlying distribution, so it is significantly sensitive to localized disturbances in small regions.

#### 2.4.4. Shannon–Fisher Plane

The use of the Shannon–Fisher plane was originally proposed by Vignat and Bercher [47], who defined it to show that through the simultaneous examination of both Shannon Entropy and Fisher Information, the non-stationary behavior of a complex signal may be characterized. Without any assumption on the nature of the data, the Shannon–Fisher area can be simply defined as:(17)D={(H,F)|0≤H≤1,0≤F≤1}

Using this plane, we can find that our system lies in a very ordered state when the Shannon Entropy H∼0 and the Fisher Information F∼1. However, when the system stays in a very disordered state, we obtain that H∼1 and F∼0 [48]. In general, it is widely accepted that the Shannon–Fisher Information plane is an effective tool to contrast global and local characteristics of a given probability distribution. In our case, and as performed by Freitas et al. [37], each of the networks is placed in the Shannon–Fisher plane that arises from the two measurements that have been explained in the previous sections.

### 2.5. Quantification Measures

We define the null hypothesis as that the population median of all of the sites are equal, the Kruskal–Wallis [49] test was used twice to quantify the magnitude of the effects of the acquisition sites for SE and FI. All possible combinations of datasets, atlases, and harmonization methods were analyzed: 21 cases with no harmonization, 21 cases harmonized with ComBat, 19 cases with CovBat, and 1 case with the traveling-subject method. It is worth noting that two cases (ABIDE/Craddock400/CovBat and ADHD-200/Craddock351/CovBat) are missing due to limitations in computing resources. As stated in a previous section, for the traveling-subject method, an extra dataset is needed, so the only possible case to evaluate that method is the one from the SRPBS dataset. The transformation $p^{\prime }=-\log p$ was applied to the *p*-values obtained with the test to facilitate the comparison and interpretation of the results.

## 3. Results

### 3.1. Quantitative Analysis

Table 1 shows the comparison of the harmonization methods for the different datasets, atlases, and Information Theory measures, while the figures in the next subsection visually illustrate the performance of each method described in this manuscript. Analyzing the results for the case without the harmonization stage, it is observed that the influence of the acquisition site is considerable, even more so in the IMPAC database. In addition, the impact is greater in the SE measure than in the FI measure for all cases and in the atlases with a larger number of ROIs for most cases (ABIDE/Dosenbach is the largest exception). When applying the ComBat method, substantial improvements are obtained in all cases, i.e., the site influences are unnoticeable and not statistically significant, achieving p≥0.05 in 12 cases. The harmonization is still better for the atlases with lower amounts of ROIs, but now the impact on the Information Theory measures is reversed: for all cases, the site effects are stronger for FI than for SE. The data for which the quality of the harmonization obtained are the lowest comes from the ADHD-200 dataset. This is due, as can be seen in Figure 2, to the existence of an outlier that may come from severe misalignments present in the scan of that particular subject. The CovBat method achieves similar results, slightly outperforming ComBat in most cases, especially when applied to atlases with no more than 200 ROIs. Finally, the traveling-subject method also represents an improvement over the case without harmonization, but it is almost insignificant compared to ComBat and CovBat.

### 3.2. Visualization

The relationships observed by analyzing Table 1 can also be visualized by plotting the Shannon–Fisher plane for the different cases. In this subsection, the most relevant results for the analysis are presented, while the others can be found in the Appendix A. The first row of Figure 3 shows the distribution of all the subjects of the IMPAC dataset within the Shannon–Fisher plane without previously applying a harmonization technique. Acquisition sites 28 and 31 present a significant distance from the center, so they are shown separately in the figures. In the second row, the planes corresponding to the ComBat method are presented, where the distribution is grouped in a much more uniform way. The same happens in the third row for the CovBat method.

In Figure 4, both the effectiveness of ComBat/CovBat and the inability of the traveling-subject method to harmonize the measures extracted from the SRPBS dataset are evident. For the unharmonized case, we can observe that sites 5 and 6 lay outside the “cloud” of the rest of the measures, which can be attributed to acquisition differences due to already mentioned factors other than the biological ones (e.g., differences in the actual equipment, differences in the used functional BOLD MRI sequence settings, improper calibration, etc.) This separation is no longer appreciable after the application of ComBat or CovBat but remains practically unchanged with the traveling-subject method.

Figure 2 and Figure 5 show the resulting planes after applying the ComBat and CovBat methods to, respectively, the ABIDE dataset and the ADHD-200 dataset. As can be seen at the bottom left of the planes corresponding to the ADHD-200 dataset, there is an outlier that could not be removed with any harmonization method, probably caused by some kind of misalignment in the scanning device.

## 4. Discussion and Conclusions

These results provide new evidence about the importance of having techniques capable of removing unwanted biases caused by the acquisition sites from rs-fMRI images in order to merge data obtained by means of different scanners without incurring methodological errors. A set of tools based on Information Theory was presented to discern the quality of harmonization techniques for multi-site rs-fMRI measurements, allowing the quality of such techniques to be verified. This set of tools, when applied to data harmonized with different techniques, makes it possible to determine if there are still traces of the original locations or if the data are reliable for the subsequent specific treatment depending on the problem to be treated. We should mention that part of inter-site heterogeneity might be caused by differences in sample demographics (e.g., race, education, background), which cannot be addressed with the harmonization methods presented here. On the other hand, we observed that the tools performed best with a low number of ROIs, as shown in the Results Section. This is probably due to the finer granularity and higher detail associated with a higher-order (i.e., with more ROIs) atlas, which allows capturing more inter-subject features (i.e., biological features that do not depend on the site). Further, variability in fMRI imaging parameters across sites may affect the quality of harmonization. In general, we observed that ABIDE, IMPAC, and ADHD-200 MRI scan parameters vary a lot from site to site. As a consequence, this could induce a lot of variability in the BOLD SNR obtainable from each site. Given that the rs-fMRI connectivity metrics have sensitive fMRI BOLD SNR, this, in turn, could have an impact on the effectiveness of harmonization. For instance, as we mentioned, the disalignment in some individual cases may force ComBat and CovBat to fail to fully harmonize the full ADHD dataset, as can be seen in Figure 2, which could be due to differences in MRI imaging parameters between that site and other sites in the ADHD dataset, and between individual runs.

In general, we observe that the theoretical information analysis reveals that ComBat and CovBat provide better results than the traveling-subject method, showing that these tools are effective in discerning the subtle details of the registration site that were not removed by the harmonization method. As a rule of thumb, we could say that CovBat provides the best results for datasets with less than 200 ROIs, while ComBat excels otherwise. The harmonization effectiveness of both Normalized Network Shannon Entropy (SE) and Normalized Network Fisher Information (FI) measures seem to deteriorate as the number of ROIs in the parcellation schemes increases. This could be due to variations between sites in the fMRI SNR. SE measures uncertainty, and thus a larger number of ROIs implies smaller sizes, which makes them more sensitive to variations in fMRI SNR and hence variability in scanning parameter-related site effects. FI, in turn, measures information, and thus a larger number of ROIs will be more sensitive to site effects as they yield more information about the site.

It is worth noticing that the results obtained in this work do not correspond to those obtained by Yamashita and co-authors [31], where they achieved a greater reduction in measurement bias using their traveling-subject method than with other techniques, i.e., ComBat. This dichotomy could be an indication that their harmonization technique is useful in specific analyses, as performed in their paper, but is not robust to other processing methodologies, such as Information Theory measures. In particular, our analysis of the original data with the Shannon–Fisher plane revealed that the site-related information was not completely removed by the harmonization process, which may render the method inadequate for further comparative analysis, while ComBat presents a more general and robust performance. Another disadvantage of the traveling-subject method is its high cost and time consumption due to the need for a large group of participants to travel to all the sites involved. Therefore, applying this method to correlation matrices requires a wider logistic basis to achieve significant harmonization, while both ComBat and CovBat present very good results.

Finally, it is worth mentioning that the workflow presented in this paper cannot be applied directly to images obtained with rs-fMRI studies; it is necessary to have the corresponding BOLD time series. Hence, for this study, from the wide spectrum of existing harmonization methods [20], we used the three compatible with this type of data. One interesting avenue for further research is to extend the workflow to more general types of information, which could be performed by properly defining the respective information theoretical measures. Further, for future work, we consider it relevant to be able to extend the analysis through the Shannon–Fisher plane to graphs and probabilistic networks (without requiring prior thresholding). Finally, we will investigate if these results can also be reproduced using the Pearson correlation matrix instead of the phase interaction matrix.

## Figures and Tables

**Figure 1 brainsci-12-01219-f001:**
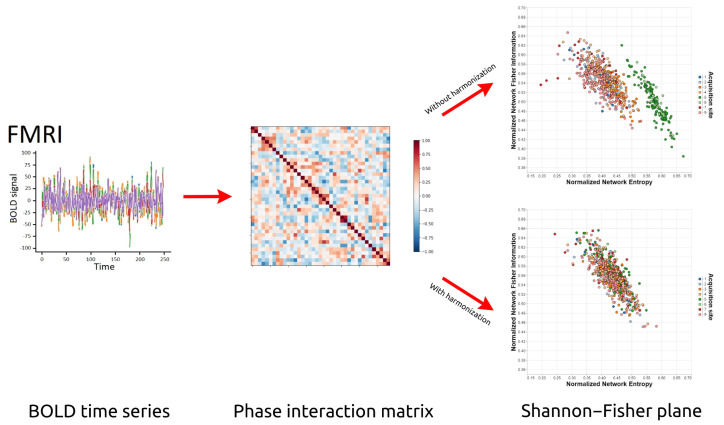
Overview of our technique to analyze, with the help of Information Theory, the effectiveness of the harmonization techniques for multi-site acquired data. From a set of multi-site fMRI data, we generate the Phase Interaction Matrix and, from there, we verify in the Shannon–Fisher plane the quality of the application of different harmonization techniques.

**Figure 2 brainsci-12-01219-f002:**
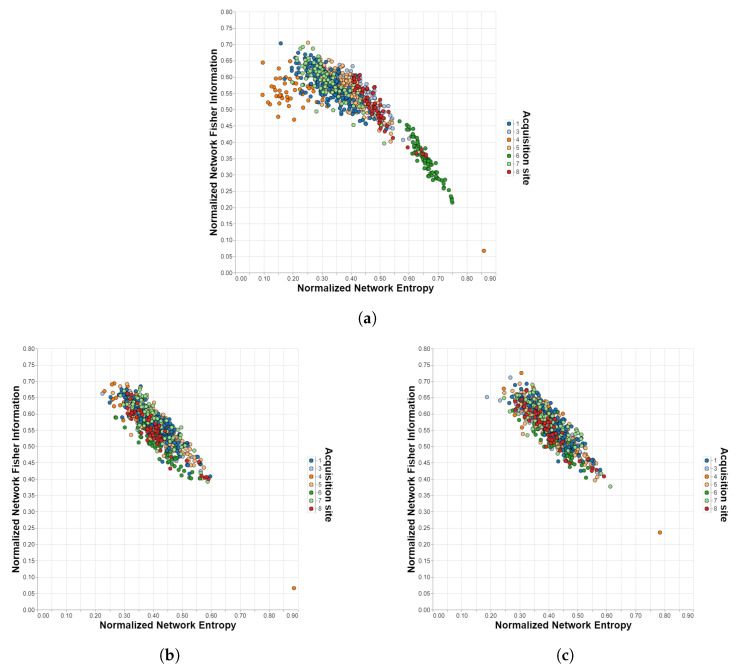
Shannon–Fisher planes for ADHD-200/Harvard-Oxford with different harmonization methods. Each point corresponds to a different subject in the dataset and each color to a different acquisition site. (**a**) Unharmonized. (**b**) ComBat. (**c**) CovBat.

**Figure 3 brainsci-12-01219-f003:**
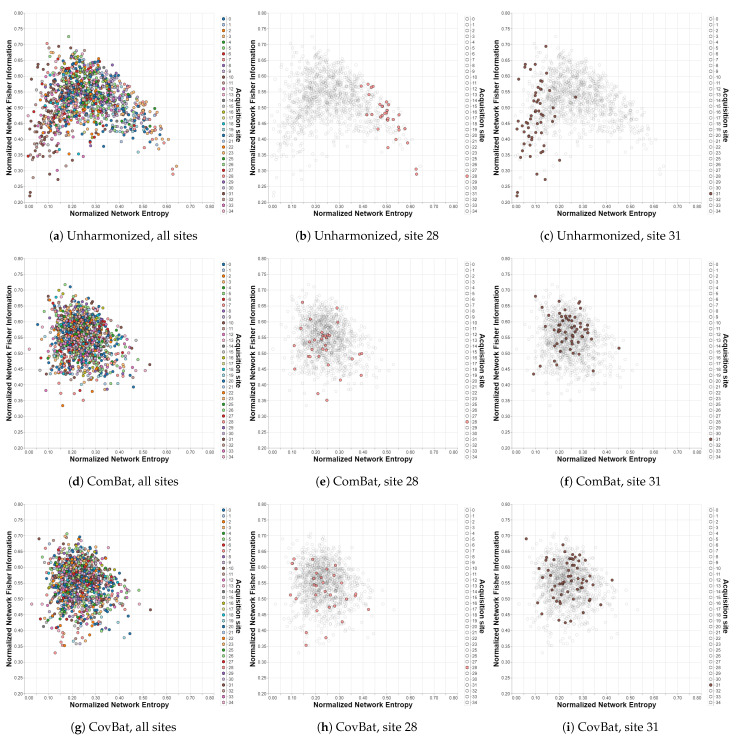
Shannon—Fisher planes for IMPAC/MSDL with different harmonization methods. Sites 28 and 31 are shown separately for each case to emphasize the impact of harmonization. Each point corresponds to a different subject in the dataset and each color to a different acquisition site. (**a**–**c**) Unharmonized. (**d**–**f**) ComBat. (**g**–**i**) CovBat.

**Figure 4 brainsci-12-01219-f004:**
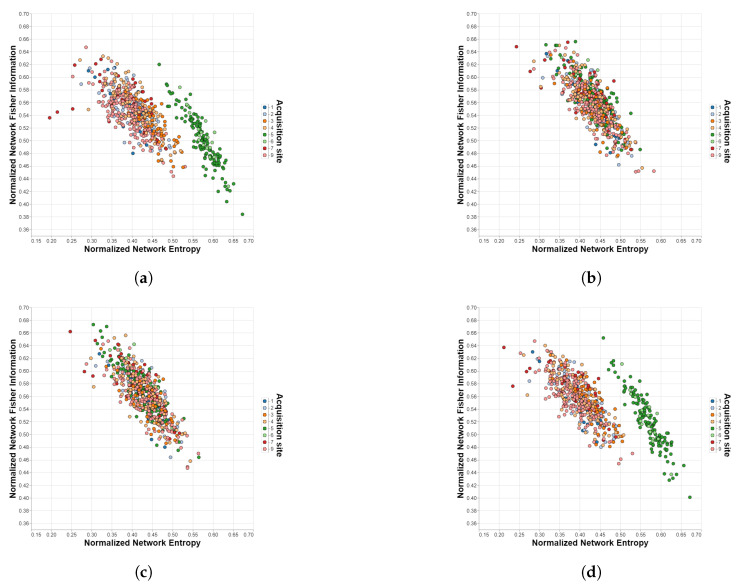
Shannon—Fisher planes for SRPBS with different harmonization methods. Each point corresponds to a different subject in the dataset and each color to a different acquisition site. (**a**) Unharmonized. (**b**) ComBat. (**c**) CovBat. (**d**) Traveling-subject.

**Figure 5 brainsci-12-01219-f005:**
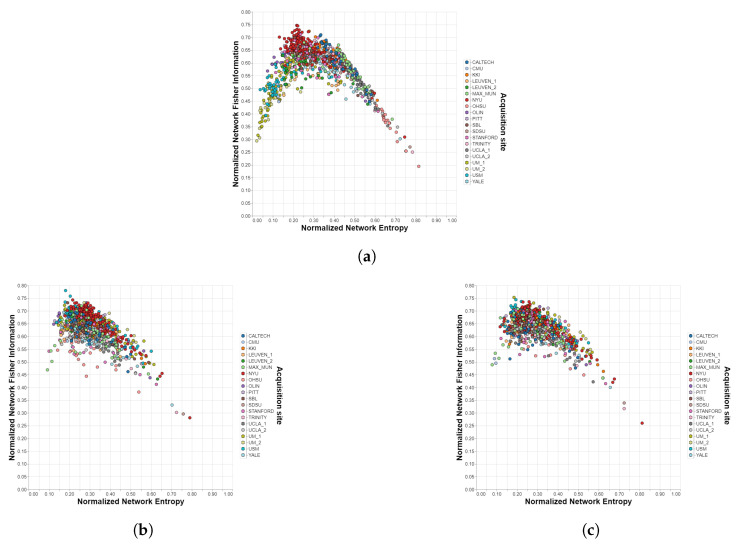
Shannon—Fisher planes for ABIDE/Dosenbach with different harmonization methods. Each point corresponds to a different subject in the dataset and each color to a different acquisition site. (**a**) Unharmonized. (**b**) ComBat. (**c**) CovBat.

**Table 1 brainsci-12-01219-t001:** Transformed *p*-values obtained with the Kruskal–Wallis test for the components of the Shannon–Fisher plane. Bold values represent $p^{\prime }\leq 1.301$, which corresponds to p≥0.05 and, therefore, evidence is not enough to reject the null hypothesis.

Dataset	Atlas	ROIs	Normalized Network Shannon Entropy				Normalized Network Fisher Information			
			Unharmonized	ComBat	CovBat	Traveling	Unharmonized	ComBat	CovBat	Traveling
IMPAC	MSDL	39	131.949	**0.242**	**0.075**	-	24.794	4.586	**1.012**	-
	Harvard-Oxford	48	137.060	**0.663**	**0.449**	-	34.342	6.264	7.326	-
	Basc064	64	131.723	**0.333**	**0.099**	-	37.705	2.725	1.860	-
	Basc122	122	149.840	**1.206**	**0.511**	-	87.554	10.588	8.960	-
	Basc197	197	159.513	2.413	1.531	-	117.111	19.124	13.914	-
	Craddock	249	166.695	3.684	4.069	-	123.781	29.584	20.274	-
	Power	264	178.503	6.156	8.997	-	136.231	53.021	37.682	-
ABIDE	Talaraich–Tournoux	97	27.416	**0.002**	**0.000**	-	12.251	**1.260**	**0.479**	-
	Harvard-Oxford	111	26.954	**0.077**	**0.000**	-	15.395	2.254	**0.645**	-
	AAL	116	29.131	**0.009**	**0.000**	-	19.490	2.355	**0.926**	-
	Eickhoff–Zilles	116	27.150	**0.009**	**0.000**	-	19.323	2.347	**0.799**	-
	Dosenbach	161	104.905	2.649	1.828	-	82.129	47.327	28.534	-
	Craddock	200	41.585	**0.258**	**0.000**	-	34.443	7.527	3.341	-
	Craddock	400	61.355	**0.682**	-	-	53.694	16.122	-	-
ADHD-200	Talaraich–Tournoux	97	100.340	8.189	4.377	-	58.882	15.115	13.039	-
	Harvard-Oxford	111	102.636	6.248	3.334	-	55.904	16.440	12.654	-
	AAL	116	102.823	8.783	5.476	-	58.034	17.995	15.119	-
	Eickhoff–Zilles	116	103.621	9.090	6.269	-	61.557	19.405	14.527	-
	Craddock	190	114.141	12.894	6.650	-	75.404	34.502	28.517	-
	Craddock	351	120.486	15.716	-	-	88.681	51.375	-	-
SRPBS	-	268	77.116	**0.285**	2.137	72.149	27.700	4.349	4.813	18.049

## Data Availability

Publicly available datasets were analyzed in this study. This data can be found in https://paris-saclay-cds.github.io/autism_challenge/, (accessed on 15 April 2022), http://preprocessed-connectomes-project.org/abide/, (accessed on 15 April 2022), http://preprocessed-connectomes-project.org/adhd200/, (accessed on 15 April 2022), and https://bicr.atr.jp/dcn/en/download/harmonization/, (accessed on 15 April 2022).

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
