# Peer review of "Assessing Multi-Site rs-fMRI-Based Connectomic Harmonization Using Information Theory"

_brainsci, 2022, doi:10.3390/brainsci12091219_

Round 1

Reviewer 1 Report

This is a generally well written paper brining new ideas to assessing harmonization techniques with Information Theory. Here are few comments after my review.

1. The introduction to Information Theory is lacking. There is no clear theoretical or heuristic account of the relevance of information theory to the subject matter of interest.

2. There are cofounding factors such as age, disease and preprocessing parameters that all need to be addressed. It would be clearer if related information can be provided in the manuscript or in supplementary materials.

3. What is the number of travelling subjects? I assume this number would affect the harmonization effect significantly.

4. The definition and interpretation of Fisher Information and Shannon-Fisher plane is inadequate.   

6. The dichotomy between the work and previous findings are not well descripted and explained.

7. Limitation of the work is not discussed. 

Author Response

This is a generally well written paper brining new ideas to assessing harmonization techniques with Information Theory. Here are few comments after my review.

  1. The introduction to Information Theory is lacking. There is no clear theoretical or heuristic account of the relevance of information theory to the subject matter of interest.

We have added, around line 62 in the Introduction section, applications of Information Theory to the specific subject of our paper, including 5 new bibliographic references that roughly cover the most recent and relevant findings in the area. Also, we have improved the explanations for the Fisher Information (Section 2.4.3) and the Shannon-Fisher plane (Section 2.4.4).

  1. There are cofounding factors such as age, disease and preprocessing parameters that all need to be addressed. It would be clearer if related information can be provided in the manuscript or in supplementary materials.

We have clarified that. In this work, the covariances of interest were defined as the gender, age, diagnosis, and site of acquisition of each of the subjects. This is now explained at the beginning of Section 2.3, where all the harmonization techniques are presented.

  1. What is the number of travelling subjects? I assume this number would affect the harmonization effect significantly.

Actually, the original travelling paper itself clarified that they had “nine healthy participants (all men; age range: 24–32 y; mean age: 27 ± 2.6 y) were scanned at each of 12 sites, which included the nine sites in the SRPBS dataset, and produced a total of 411 scan sessions”. This is now described in Section 2.1.5, where the traveling-subject dataset is described.

  1. The definition and interpretation of Fisher Information and Shannon-Fisher plane is inadequate.   

In Sections2.4.3 and  2.4.4 we added more detail, including a precise formulation and the bibliographic references to where this information theoretical measure was first introduced and its meaning and formulation was thoroughly explained. 

  1. The dichotomy between the work and previous findings are not well descripted and explained.

The Discussion section has been updated to include a detailed description of this point, around line 292 of the current document. In particular, we have explained that this dichotomy could be an indication that their harmonization technique is useful in specific analyses, as done in their paper, but is not robust to other processing methodologies such as Information Theory measures. In particular, our analysis of the original data with the Shannon-Fisher plane revealed that the site-related information was not completely removed by the harmonization process, which may render the method inadequate for further comparative analysis, while ComBat presents a more general and robust performance.

  1. Limitation of the work is not discussed.

We have updated our Discussion and conclusions Section to mention that the workflow presented in this paper cannot be applied directly to images obtained with rs-fMRI studies, it is necessary to have the corresponding BOLD time series. Hence, for this study, from the wide spectrum of existing harmonization methods, we used the three compatible with this type of data. One interesting avenue for further research is to extend the workflow to more general types of information, which could be done by properly defining the respective information theoretical measures. Also, for future work, we consider it relevant to be able to extend the analysis through the Shannon-Fisher plane to graphs and probabilistic networks (without requiring prior thresholding).

Reviewer 2 Report

The main question of this paper is about the harmonization of the result of rs-fMRI from different centers. Analyzing multi-center data is always one of the most important issues in biostatistics and researchers have faced more problems with MRI data than the other type of medical data. The originality of analyzing is good however the writers gave more information on the method parts which are not covered in the result. In the paper, there was not any information about the demographic variables which could affect the analysis. Moreover, when different analyzing methods were used, it is better to discuss which method is appreciated. It should be discussed more in the conclusion about the result of different methods.

In this paper, some harmonization methods were described. However, the result did not cover the methods exactly. It is not clear which method is appreciated?

Affiliations should be written in English. 

Author Response

The main question of this paper is about the harmonization of the result of rs-fMRI from different centers. Analyzing multi-center data is always one of the most important issues in biostatistics and researchers have faced more problems with MRI data than the other type of medical data. The originality of analyzing is good however the writers gave more information on the method parts which are not covered in the result. 

In the paper, there was not any information about the demographic variables which could affect the analysis. 

As mentioned above in reply to the Reviewer 1, covariances of interest were defined as the gender, age, diagnosis, and site of acquisition of each of the subjects. This explanation has been added to Section 2.3, where all the harmonization techniques are presented.

Moreover, when different analyzing methods were used, it is better to discuss which method is appreciated. 

We have added, at each figure caption in the manuscript, and for each result discussed, the exact combination of database and harmonization method used. We hope this change will completely clarify any doubt in this respect.

It should be discussed more in the conclusion about the result of different methods.

We expanded the Discussion section with commentaries on the origins of inter-site heterogeneity, which might be caused by differences in sample demographics (e.g., race, education, background), and that cannot be addressed with the harmonization methods presented here. Also, we clarified that we observed that the tools performed best with a low number of ROIs, as shown in the Results Section. This is probably due to the finer granularity and higher detail associated with a higher-order (i.e., with more ROIs) atlas, that allows capturing more inter-subject features (i.e., biological features that do not depend on the site).

On the other hand, we have added a comment stating that, in general, we observe that the information theoretical analysis reveals that ComBat and CovBat provide better results than the traveling-subject method, showing that these tools are effective to discern the subtle details of the registration site that were not removed by the harmonization method. We commented that, as a rule of thumb, we could say that CovBat provides the best results for datasets with less than 200 ROIs, while ComBat excels otherwise.

Finally, as commented above, we clarified which dichotomy we were referring to in our original submission.

  1. In this paper, some harmonization methods were described. However, the result did not cover the methods exactly. It is not clear which method is appreciated?

As mentioned just above, we have added this clarification to every instance of analysis for every dataset and method used.

  1. Affiliations should be written in English.

This has been fixed, thanks for pointing this out!

Reviewer 3 Report

This study compares three approaches to “fMRI harmonisation” (the practice of correcting differences across acquisition sites/samples).  The authors included four multi-site databases, each one with various atlases applied. Their key outcomes were two information-theoretic measures, Shannon entropy and Fisher information, which they used to test whether the harmonisation was improved. They found that two of the three techniques were effective except in one of the datasets. The article is scientifically sound and has some novelty/value to the field, but the results and discussion need further additions, which I have suggested below. The article’s presentation is good but editing for English is needed.

Abstract

·        The authors should amend the title and abstract to clarify that this article is about harmonisation of fMRI connectomics/networks (rather than task-based or other analyses).

·        “Identifiable” has a specific meaning pertaining to data privacy, so I would edit this sentence to “since the acquisition site can still be determined from the fMRI data after the processing” or similar.

Introduction

·        Why mention in the opening sentence that MRI has a high spatial resolution? This seems irrelevant to the paper and, in fact, the resolution of fMRI data is not considered high compared to most sequences and other imaging modalities.

·        Worth mentioning that part of inter-site heterogeneity might be the difference in sample demographics (e.g. race).

·        Figure 1 “BOLD time series” uses a drawn style which is inconsistent with the rest of the figure. Secondly, the x-axis shows the ROI number instead of the (more intuitive) “time”, although it is not technically wrong.

Methods

·        The authors should clarify that, since the aim is to compare across the sites within each multi-site dataset, it does not matter that each downloaded dataset already had different processing steps already applied. Secondly, that three datasets were downloaded as sets of timeseries, already split according to region of interest, while one was downloaded as a connectivity matrix of Pearson correlation values.

·        The authors should clarify in the SRPBS data section that it included travelling-subject data, and is thus applicable to the travelling subject harmonisation methodology.

Results

·        Table 1: suggest changing the “-“ column header to “unharmonised”. Suggest changing “Shannon” and “Fisher” to “Shannon entropy” and “Fisher information”. Suggest adding a sentence in the table legend to state to an uninformed reader what a significant p-value here represents (no effect of the acquisition site).

·        Please join Figures 2, 3 and 4 together into a single figure and add subheadings to each row e.g. “unharmonised”, “ComBat”, “CovBat” (as in Figure 5).

·        Please show the S-F planes for the other datasets in Figure 5. Perhaps the Figure could be condensed to allow four rows – one for each of the databases with four columns of graphs. I believe this is important because, as we can see from Figure 5, most sites are already overlapping closely with the exception of two sites (green, light green). This raises the question, “to what extent are the data unharmonised in the first place and how does that affect harmonisation?” Was the ADHD-200 database more difficult to harmonise because its sites had a larger spread on the S-F plane? This could add another dimension to the results and might explain some of the findings.

Discussion

·        In general the discussion is too thin and leaves many unanswered questions.

·        Which method performed best and worst, and why?

·        Which harmonisation methods were not evaluated here?

·        Why does the ROI number influence the results?

·        Why were sites 5 and 6 in SRPBS so different from the others, and how did it impact the results (Figure 5)?

·        Why did the ADHD dataset fail to achieve significant harmonisation with any approach?

·        How complete is the set of relevant harmonisation methods that were compared? Are any missing?

·        What recommendations do the authors have for researchers needing to harmonise their fMRI connectomes regarding the choice of atlas, number of sites involved, harmonisation method, etc?

·        What are the limitations of (a) the respective harmonisation methods tested and (b) this study?

Author Response

This study compares three approaches to “fMRI harmonisation” (the practice of correcting differences across acquisition sites/samples).  The authors included four multi-site databases, each one with various atlases applied. Their key outcomes were two information-theoretic measures, Shannon entropy and Fisher information, which they used to test whether the harmonisation was improved. They found that two of the three techniques were effective except in one of the datasets. The article is scientifically sound and has some novelty/value to the field, but the results and discussion need further additions, which I have suggested below. The article’s presentation is good but editing for English is needed.

Abstract

  1. The authors should amend the title and abstract to clarify that this article is about harmonisation of fMRI connectomics/networks (rather than task-based or other analyses).

We have updated the title to include the word “Connectomics”, which we believe would help clarify this point and make the paper much easier to read and classify for potential readers. Thanks for the suggestion!

  1. “Identifiable” has a specific meaning pertaining to data privacy, so I would edit this sentence to “since the acquisition site can still be determined from the fMRI data after the processing” or similar.

We have edited the sentence as suggested (at the end of the Abstract, line 12). Thanks for the comment!

Introduction

  1. Why mention in the opening sentence that MRI has a high spatial resolution? This seems irrelevant to the paper and, in fact, the resolution of fMRI data is not considered high compared to most sequences and other imaging modalities.

    This comment has been removed. Thanks for noticing!

  1. Worth mentioning that part of inter-site heterogeneity might be the difference in sample demographics (e.g. race).

    This was addressed in the replies to the two other reviewers. To address this point, as already mentioned above, we have clarified that the covariances of interest were defined as the gender, age, diagnosis, and site of acquisition of each of the subjects. This is now explained at the beginning of Section 2.3, where all the harmonization techniques are presented.

  1. Figure 1 “BOLD time series” uses a drawn style which is inconsistent with the rest of the figure. Secondly, the x-axis shows the ROI number instead of the (more intuitive) “time”, although it is not technically wrong.

    We have updated the image so that the style is consistent, as well as correcting the labels, which were mistaken. Thanks for the suggestion!

Methods

  1. The authors should clarify that, since the aim is to compare across the sites within each multi-site dataset, it does not matter that each downloaded dataset already had different processing steps already applied. Secondly, that three datasets were downloaded as sets of timeseries, already split according to region of interest, while one was downloaded as a connectivity matrix of Pearson correlation values.

As is now explained at the beginning of Section 2.1, iwe describe the different datasets used for testing. Since the aim is to compare the sites within each multi-site dataset, we can see that it does not matter that each downloaded dataset had different processing steps already applied. Also, it is worth noting that three of the datasets are given as sets of time series (IMPAC, ABIDE, and ADHD-200), while one was given as a set of connectivity matrices of Pearson correlation values (SRPBS).

  1. The authors should clarify in the SRPBS data section that it included travelling-subject data, and is thus applicable to the travelling subject harmonisation methodology.

    As suggested, we have added this comment at the end of the respective Section, 2.1.4. Thanks!

Results

  1. Table 1: suggest changing the “-“ column header to “unharmonised”. Suggest changing “Shannon” and “Fisher” to “Shannon entropy” and “Fisher information”. Suggest adding a sentence in the table legend to state to an uninformed reader what a significant p-value here represents (no effect of the acquisition site).

    We have updated the table as suggested. Thanks for the comment!

  1. Please join Figures 2, 3 and 4 together into a single figure and add subheadings to each row e.g. “unharmonised”, “ComBat”, “CovBat” (as in Figure 5).

    We have unified the figures, and added new ones as suggested. Now the paper looks more complete, thanks!

  1. Please show the S-F planes for the other datasets in Figure 5. Perhaps the Figure could be condensed to allow four rows – one for each of the databases with four columns of graphs. I believe this is important because, as we can see from Figure 5, most sites are already overlapping closely with the exception of two sites (green, light green). This raises the question, “to what extent are the data unharmonised in the first place and how does that affect harmonisation?” Was the ADHD-200 database more difficult to harmonise because its sites had a larger spread on the S-F plane? This could add another dimension to the results and might explain some of the findings.

    We have added figures with the datasets ABIDE: Dosenbach and ADHD-200: Harvard-Oxford, as the IMPAC dataset was already in the original figures. In any case, we would like to emphasize that the figures were already present in the Supplemental Material, we simply decided not to include them because of space limitations. However, we feel now the main part of the manuscript reads better and is more complete, so thanks for the suggestion!

Discussion

In general the discussion is too thin and leaves many unanswered questions.

    We have largely expanded the Discussion Section with answers to each of these and other comments and suggestions, as well as other aspects  which we realized were not properly commented on, please see below. Thanks for the advice!

11.Which method performed best and worst, and why?

    We enlarged the Discussion Section, among other things explaining that, in general, we observe that the information theoretical analysis reveals that ComBat and CovBat provide better results than the traveling-subject method, showing that these tools are effective to discern the subtle details of the registration site that were not removed by the harmonization method. Also, we have clarified that, as a rule of thumb, we could say that CovBat provides the best results for datasets with less than 200 ROIs, while ComBat excels otherwise.

  1. Which harmonisation methods were not evaluated here?

    Also in the conclusions section, we mention that, in our opinion, one interesting avenue for further research is to extend the workflow to more datasets, techniques, and more general types of information, which could be done by properly defining the respective information theoretical measures. For instance, the techniques described in reference [20] present a wide spectrum of existing harmonization methods.

  1. Why does the ROI number influence the results?

    In the Discussion again we commented that we observed that the tools performed best with a low number of ROIs, as shown in the Results Section. This is probably due to the finer granularity and higher detail associated with a higher-order (i.e., with more ROIs) atlas, that allows capturing more inter-subject features (i.e., biological features that do not depend on the site). This is on line 281 in the manuscript.

14.Why were sites 5 and 6 in SRPBS so different from the others, and how did it impact the results (Figure 5)?

    In the Results Section, we have clarified that, for the unharmonized case, we can observe that sites 5 and 6 lay outside the "cloud" of the rest of the measures, which we attribute to some kind of miscalibration of the devices. This separation is no longer appreciated after the application of ComBat or CovBat, but remains practically unchanged with the traveling-subject method.

  1. Why did the ADHD dataset fail to achieve significant harmonisation with any approach?

    This is a good question, that we think is associated with the outliers in the dataset. As can be seen at the bottom left of the planes corresponding to the ADHD-200 dataset, there is an outlier that could not be removed with any harmonization method, probably caused by some kind of misalignment in the scanning device. This has been clarified in the Results Section, on line 265.

  1. How complete is the set of relevant harmonisation methods that were compared? Are any missing?

    This is related to question 12, above, and the answer is that, although there are missing methods, we have compared the ones most relevant to the kind of connectomic data we were dealing with, and thus the necessary update in the title. There are other harmonization techniques that could be used, given a proper definition of the required Information Theory tools.

  1. What recommendations do the authors have for researchers needing to harmonise their fMRI connectomes regarding the choice of atlas, number of sites involved, harmonisation method, etc?

    Following our answer to the question about which method provided better results, we added that, as a rule of thumb, we could say that CovBat provides the best results for datasets with less than 200 ROIs, while ComBat excels otherwise.

  1. What are the limitations of (a) the respective harmonisation methods tested and (b) this study? 

As mentioned previously in our answer to Reviewer 1, we have updated our Discussion and conclusions Section to mention that the workflow presented in this paper cannot be applied directly to images obtained with rs-fMRI studies, it is necessary to have the corresponding BOLD time series. Thus, one interesting avenue for further research is to extend the workflow to more general types of information, which could be done by properly defining the respective information theoretical measures. Also, for future work, we consider it relevant to be able to extend the analysis through the Shannon-Fisher plane to graphs and probabilistic networks (without requiring prior thresholding).

Round 2

Reviewer 2 Report

The manuscript was updated very well. 

Author Response

Thanks for your positive comments.

Reviewer 3 Report

The authors have addressed my questions sufficiently.

Author Response

Thanks for your positive comments.